# Investigation on the Influence of Flow Passage Structure on the Performance of Bionic Pumps

Ertian Hua [1,2,*], Haitao Luo [1], Rongsheng Xie [1,3], Wanqian Chen [1], Shouwei Tang [1] and Dongyang Jin [1]

1    College of Mechanical Engineering, Zhejiang University of Technology, Hangzhou 310003, China
2    Zhejiang Engineering Research Center for Advanced Hydraulic Equipment, Hangzhou 310018, China
3    School of Mechanical & Automotive Engineering, Zhejiang University of Water Resources and Electric Power, Hangzhou 310018, China
*    Correspondence: het@zjut.edu.cn; Tel.: +86-135-8811-4369

**Abstract:** The flapping hydrofoil bionic pump drives the hydrofoil to make simple harmonic motion and completes one-way water pumping in the flow passage. As a new pump device that can realize ultra-low head water delivery, the flapping hydrofoil device can effectively enrich the drainage methods of plain rivers and improve water delivery efficiency, and the passage structure is the key factor of ultra-low head devices. In this paper, the two-dimensional flow passage models are established, and the flapping of the airfoil is realized by using the dynamic grid technology. Based on the continuity equation, k-ε turbulence model, and Reynolds time-averaged equation, the flapping hydrofoil device is simulated by transient calculation. The hydraulic performance characteristics of various passages with different widths, such as square passages, micro-arc passages, and convergent–divergent passages, are calculated and simulated. The results show that, under the fixed motion parameters, the narrower the passage width, the higher the outlet velocity, lift, and efficiency of the device, the lower the flow rate. The contraction–expansion pipe can effectively improve the efficiency and flow rate of the device, and, before the wake is stable, the longer the contraction section the better the lifting effect. However, the micro-arc pipeline will affect the formation of a double-row anti-Karman vortex street, resulting in greater energy loss and in its hydraulic performance being inferior to that of the square passage.

**Keywords:** flapping hydrofoil; hydrodynamic performance; numerical simulation; test verification

## 1. Introduction

In plain river network areas, the terrain is low and the water mobility is poor, resulting in low drainage and flood discharge capacity, serious water pollution, and other problems that are difficult to solve [1]. It is an important idea to improve the water environment quality through [2,3] pump sluice joint dispatching to improve the hydrodynamic force. The existing research [4] shows that increasing the flow rate of the river is conducive to the degradation of pollutants, thus improving the self-purification capacity of the water body. However, the pump station has problems such as low operation efficiency and poor stability [5–7] under the ultra-low head, which cannot adapt to the situation where the head of a plain small river is almost zero and will destroy the original ecology of the channel [8]. As a water-pushing device, the flapping hydrofoil bionic pump [9,10] has the characteristics of a simple structure, low construction cost, easy erection, and head close to zero. It can effectively enrich the drainage methods of plain rivers and improve the water conveyance efficiency, which is of great research value.

Flapping hydrofoil is simplified from tuna swimming tail fin movement. As early as 1909, Knoller R., it has been found through experiments that the airfoil in sinusoidal undulating motion in the steady incoming flow will generate an effective angle of attack with the incoming flow so that its normal force can generate a component force in the

positive direction [11]. After that, Von K and Burgers [12] first explained the principle of drag and lift generation of flapping hydrofoil theoretically by observing the wake structure of the two-dimensional flow. Liu [13] launched the three-dimensional numerical simulation of the bionic flapping wing and discussed the relationship between the hydrodynamic performance of the flapping wing and the direction, interconnection, and dissipation rate of the vortex ring. Triantafyllou and Anderson et al. [14,15] carried out a series of hydrofoil propulsion experiments in the MIT Towing Pool Laboratory, measured the lift drag coefficient and propulsion efficiency of the hydrofoil in the combination of undulating and pitching motions, and found that the optimal propulsion efficiency of hydrofoil propulsion is between the Strouhal number St = 0.25~0.35, and when St is high, the unsmooth change of the effective angle of attack will cause the reduction of the hydrofoil performance. Therefore, a way to improve the change of effective angle of attack by adding high-frequency terms in the undulating motion of the hydrofoil is proposed to improve the propulsion performance of flapping hydrofoil at high St. Using the structure and wave pattern similar to tuna, X Chang et al. [16] conducted a numerical study on the two turbulence models and compared them with laminar flow. The results show that the propulsion performance of the model is better at a higher Reynolds number; in addition, the numerical analysis shows that although the "thrust" of the lunar tail is relatively small, its thrust efficiency is the highest. The main reason is that the lateral energy loss is small. Boudin, A. et al. [17] carried out a numerical study of the two-dimensional rigid wing by changing the motion trajectory parameters. The study shows that the non-sinusoidal trajectory can change the flow vorticity and wake structure. Under the optimal condition, the thrust increases by 110%. Garg, N. [18] used experimental measurements to verify the optimization results of the NACA0009 hydrofoil, ensuring the optimization method of its hydraulic structure design.

The research on flapping hydrofoil mainly focuses on the motion parameters and airfoil structure. Because most of them are applied to energy acquisition and underwater thrusters, the open flow field is adopted in the numerical simulation process, without considering the wall effect. In this paper, a bionic pump is innovatively proposed by combining the flapping wing with closed passage to solve the hydrodynamic problem of plain river network. In order to further improve the pumping performance, its passage structure is studied in depth. Tang [19] pointed out that for low-head pump stations, the proportion of pipeline loss is large, and the high-efficiency section is offset, which leads to the low efficiency of the pump stations selected for the project. Xie et al. [20] effectively reduced the hydraulic loss and improved the external characteristics of the pump device by adjusting the profile of the shaft and straight pipe flow passage. Similarly, Dahmani, F et al. [21] simulated the unsteady turbulent flow of the energy collector based on the two-dimensional naca0015 series hydrofoil with different AR ratios. The results show that compared with the traditional open passage, the contraction pipe increases the incoming flow velocity, and positively affects the interaction time between the hydrofoil and the eddy current, resulting in greater vertical hydrodynamic force, thus improving the power extraction. In this paper, based on the previous research, aiming at the influence of passage width and various passage structures on the water-pushing performance of a flapping hydrofoil, the two-dimensional numerical simulation results are verified using the flapping hydrofoil device model test. The internal flow field and hydrodynamic performance of flapping hydrofoil devices in several flow passage structures are analyzed. This paper has a certain reference significance for the actual engineering design of flapping hydrofoil bionic pumps and the hydrodynamic performance research of straight pipe flow passages and contraction expansion flow passages as operation occasions.

## 2. Materials and Methods

### 2.1. Numerical Simulation Model

#### 2.1.1. Numerical Calculation Model

The main working component of the flapping wing bionic pump studied in this paper is NACA0012 airfoil, and its profile is shown in Figure 1. Chord length of c = 300 mm was

selected as the profile of its main working parts. The motion of flapping hydrofoil can be considered as the coupling of the pitching motion around the pitching axis and the heaving motion perpendicular to the water flow direction, as shown in Figure 2.

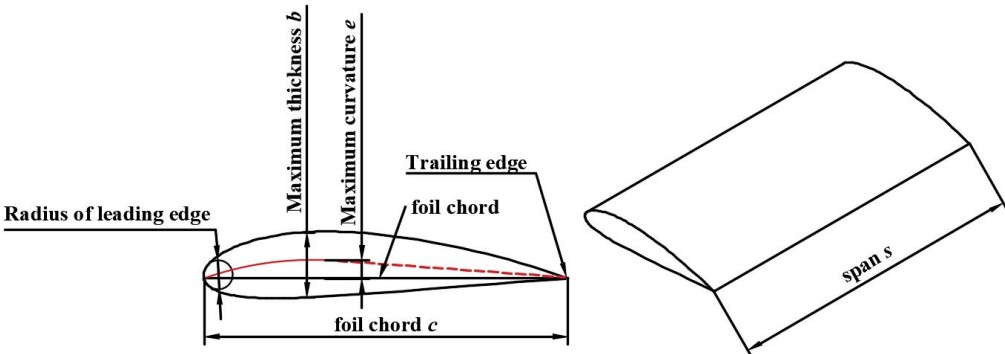

**Figure 1.** Schematic diagram of hydrofoil.

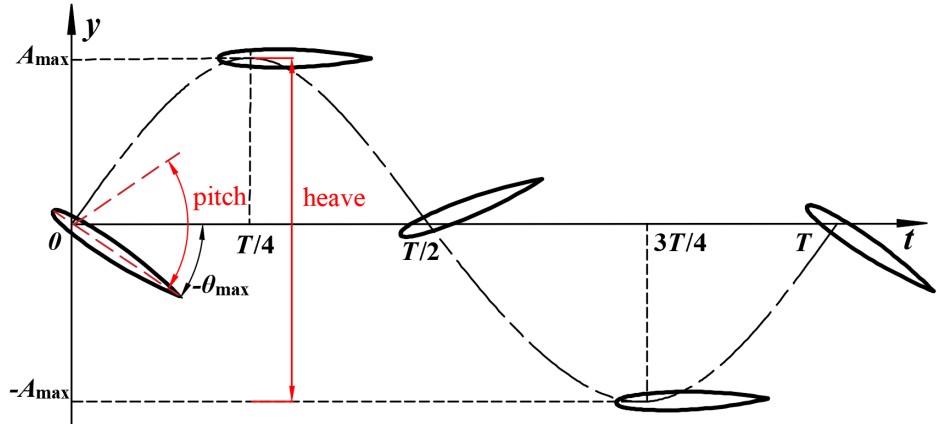

**Figure 2.** Schematic diagram of flapping hydrofoil movement.

where $A_{\max}$ represents the heaving amplitude of the flapping hydrofoil; $\theta_{\max}$ represents the pitch amplitude of the flapping hydrofoil; $T$ represents the period of motion. The basic motion equation of a flapping hydrofoil is defined by:

$$\begin{cases} y(t) = A_{\max} \sin(\omega t) \\ \theta(t) = \theta_{\max} \sin(\omega t + \phi), \end{cases} \tag{1}$$

where $y(t)$ is the heave displacement of the flapping hydrofoil, $\theta(t)$ is the pitch displacement of the flapping hydrofoil, $\omega$ is the angular frequency of flapping, $\phi$ is the phase angle between heave and pitch. Take the derivative of Equation (1) to obtain the flapping wing speed at any time:

$$\begin{cases} \dot{y}(t) = \omega A_{\max} \cos(\omega t) \\ \dot{\theta}(t) = \omega \theta_{\max} \cos(\omega t + \phi), \end{cases} \tag{2}$$

where $\dot{y}(t)$ is the heave speed of the flapping wing, $\dot{\theta}(t)$ is the pitch speed of the flapping wing.

In this paper, the fixed heave amplitude $A_{\max} = 0.5c$, pitch amplitude $\theta_{\max} = \pi/6$, phase angle $\phi = -\pi/2$, the distance from the rotating center of the flapping hydrofoil to the leading edge $l = 0.2c$, and flapping frequency $f = 1$ Hz, define the Strouhal number $St = 2fA_{\max}/\overline{U}$, which $\overline{U}$ is the average value of the outlet velocity after the flow is stabilized.

The width of the flow passage in a flapping hydrofoil device is only slightly wider than the blade span, and the airfoil and its motion do not change with the span direction, so it can be simplified as a 2D model for simulation research. To study the influence of the flow passage on the hydraulic performance of the device, the flow passage was optimized in size and profile, and the optimal flow passage design scheme was selected through comparison. The flapping hydrofoil device under different flow passage schemes is shown in Figure 3.

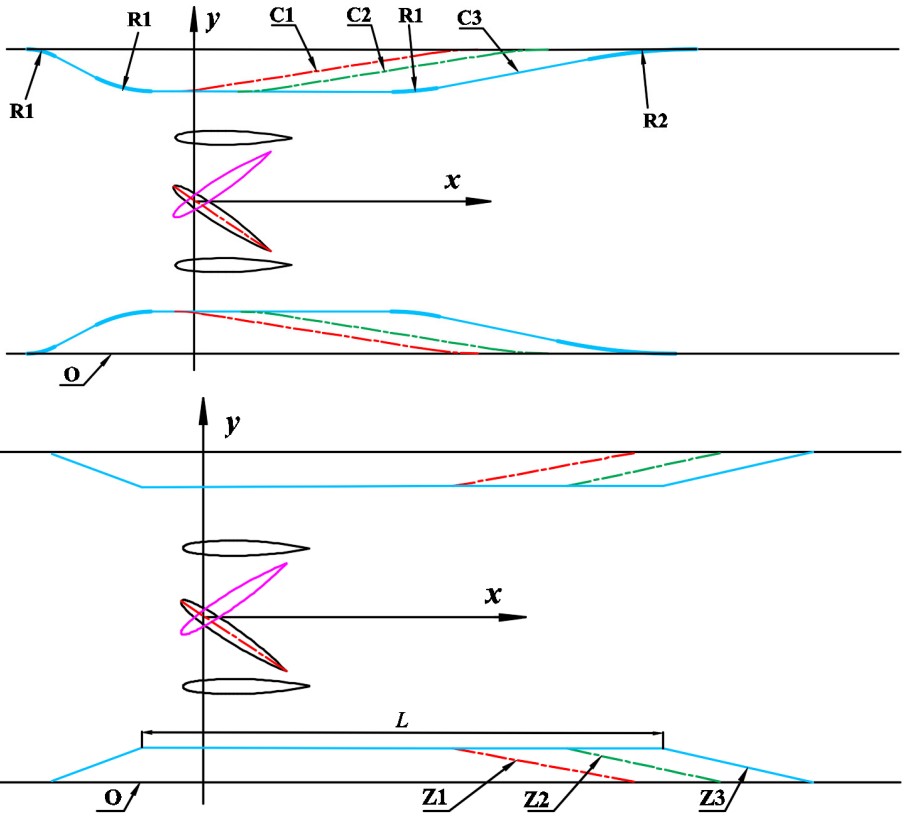

**Figure 3.** Schematic diagram of different flow passage devices.

In Figure 3, the difference between the C-type passage and Z-type passage is that all parts of the C-type passage are connected by fillet transition, while the Z-type passage adopts right angle transition. The width of the O-type rectangular flow passage is 0.8 m, the unilateral indentation width of other flow passages is 0.1 m, the distance from the contraction inlet to the hydrofoil rotation center is 0.645 m, the length of the contraction flow passage at the inlet is 0.5 m, the length of the expansion flow passage at the outlet is 0.95 m, the radius of the fillet R1 is 1 m, and the radius of the fillet R2 is 2 m. The length L of the reduced flow passage at the center of each flow passage is different, as shown in Table 1.

**Table 1.** Table Passage structure parameter table.

| Passage Type | C1 | C2 | C3 | Z1 | Z2 | Z3 |
|---|---|---|---|---|---|---|
| Length L/m | 0 | 0.5 | 1 | 1 | 1.5 | 2.0 |

### 2.1.2. Motion Parameters and Mechanical Models

In the research of flapping hydrofoils, the instantaneous thrust coefficient and the instantaneous lift coefficient are the key parameters to measure the hydrodynamic performance of a flapping hydrofoil. The calculation formulas are:

$$\begin{cases} C_x(t) = \frac{2F_x(t)}{\rho \overline{U}^2 cs} \\ C_y(t) = \frac{2F_y(t)}{\rho \overline{U}^2 cs} \end{cases} \tag{3}$$

where $F_x(t)$ is the horizontal instantaneous thrust, $F_y(t)$ is the vertical instantaneous lift, $\rho$ is the fluid density, $s$ is the span of the airfoil.

To characterize the water-pushing performance of a flapping hydrofoil, it was necessary to calculate the flow, head, and efficiency of a flapping hydrofoil hydrodynamic device.

The flow was obtained by multiplying the average velocity of the outlet section with the area, while the average head is obtained by converting the pressure difference between the inlet and outlet. The formulas are defined as follows:

$$\overline{Q} = \overline{U}sb \tag{4}$$

$$\overline{H} = \frac{\Delta \overline{p}}{\rho g} \tag{5}$$

where $Q$ is the average value of flow, $b$ is the width of the passage, taken as 0.8 m.

The average input power of flapping wing motion is calculated by

$$\overline{P} = \frac{1}{T} \left( \int_0^T F_y(t)\dot{y}(t)dt + \int_0^T M(t)\dot{\theta}(t)dt \right) \tag{6}$$

where $\overline{P}$ is the average input power of flapping wing movement, $F_y(t)$ is the vertical instantaneous lift, $M(t)$ is the instantaneous torque of airfoil around the shaft.

The average power obtained by the fluid is

$$\overline{P_1} = \Delta \overline{p} \cdot \overline{Q} \tag{7}$$

Relations (9) and (10), $\eta$ is given by

$$\eta = \frac{\overline{P_1}}{\overline{P}} \tag{8}$$

*2.2. Numerical Method*

2.2.1. Control Equation and Turbulence Model

In this paper, the commercial CFD software FLUENT was used for numerical calculation. Considering the two-dimensional incompressible turbulent flow problem, its motion control equation can be expressed as [22]

$$\frac{\partial u_i}{\partial x_i} = 0 \tag{9}$$

$$\frac{\partial u_i}{\partial t} + u_j \frac{\partial u_i}{\partial x_j} = -\frac{\partial p}{\partial x_i} + \frac{\partial}{\partial x_j}[(\gamma + \gamma_t)(\frac{\partial u_i}{\partial x_j} + \frac{\partial u_j}{\partial x_i})] \tag{10}$$

where $u_i(i = 1,2)$ is the fluid velocity, $x_i(i = 1,2)$ is the space coordinate, $P$ is the fluid pressure, $\gamma$ is the kinematic viscosity coefficient, $\gamma_t = c_\mu k^2 \varepsilon$ is the turbulent viscosity coefficient, $k$ is the turbulent kinetic energy, $\varepsilon$ is the turbulent energy dissipation rate, $c_\mu$ is the constant.

To clearly capture the vortex generated and shed by flapping hydrofoil in the flow field, this paper used the Realizable turbulence model to solve the N-S equation model. See the literature [23] for the corresponding equations.

2.2.2. Calculation Method and Boundary Conditions

ANSYS FLUENT software was used for numerical calculations to conduct transient calculations. Dynamic mesh technology was used to solve the problem of hydrofoil motion,

in which the elastic smoothing method and local reconstruction method were comprehensively used for the dynamic mesh model. For the elastic smoothing method, we set the elastic coefficient to 0.8 to reduce its impact on the grid in the far area and set the boundary point relaxation factor to 0.0006 to ensure that the grid node distribution at the boundary was not affected. For the local reconstruction method, we set the maximum mesh distortion ratio to 0.7 and the appropriate minimum mesh length standard to obtain the dynamic mesh at different times, as shown in Figure 4; the passage type is "o", with width is 0.6 m, four main points represents the four moments of flapping hydrofoil, namely, upward attack—balanced position—downward attack—balanced position. The fluid medium in the calculation domain was set as water, the flow passage and airfoil surface were set as nonslip walls, and the interface between the three calculation domains was set as Interface. The differential equations governing the fluid flow were solved by setting the Coupled algorithm to couple the pressure field with the velocity field, and the second-order upwind scheme was used for discrete time. The step size should be less than the ratio of the minimum grid scale to the flow velocity, which should be adjusted according to the calculation under different conditions.

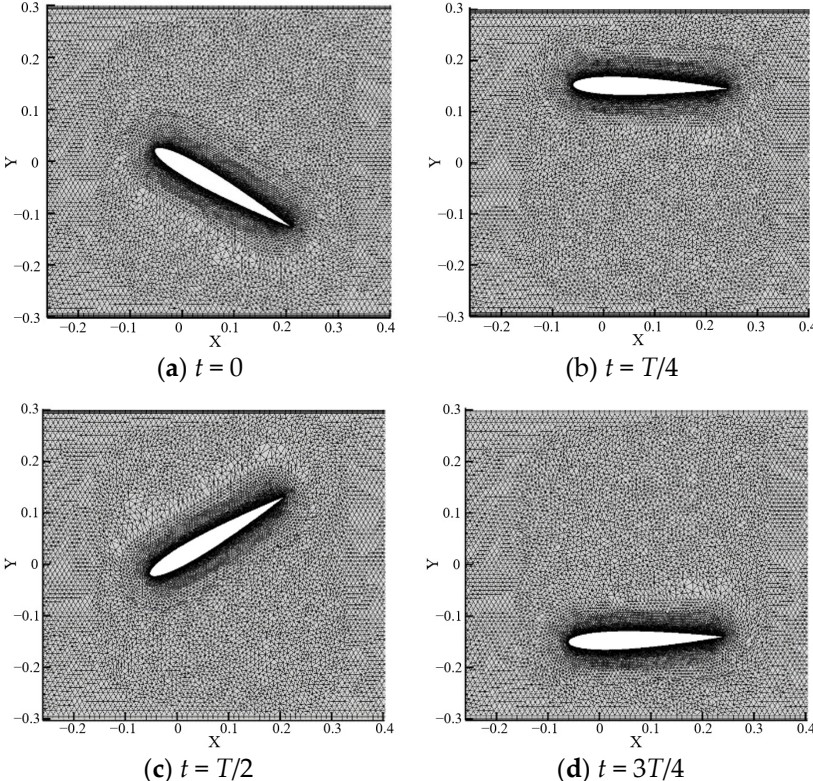

**Figure 4.** Dynamic mesh updating at four main points in a cycle.

### 2.2.3. Solver Validation

In order to verify the effectiveness of this numerical simulation method, according to the experimental research on the propulsion performance of flapping hydrofoil in the literature [24], this section has established a corresponding simulation model to simulate the propulsion performance of flapping hydrofoil and compared the simulation results with the experimental results.

The NACA0012 airfoil was selected as the working part according to the literature [24]. The chord length of the airfoil is c = 0.1 m. The dimensionless dimensions of the calculation domain are set to be (20 c, 15 c). The rotation center of the flapping hydrofoil is set at 1/3 c from the leading edge. The distance between the rotation center of the flapping hydrofoil and the entrance boundary is 5 c. In addition, the inlet velocity $U = 0.4$ m/s at the inlet boundary is set, the heaving amplitude of flapping hydrofoil $A_{\max} = 0.1$ m, and

the maximum angle of attack $a_m = 15°$, the phase angle between deep motion and pitching motion $\phi = 90°$, Reynolds number $Re = 4 \times 10^4$, and the amplitude of pitching motion of flapping hydrofoil are determined by the maximum angle of attack $a_m$ and Strouhal number $St$. At maximum angle of attack $a_m$ is the same. The simulation calculation was carried out for various working conditions with different Strouhal number $St$, and the numerical calculation results were compared with the experimental data in the literature [24]. The results are shown in Figure 5.

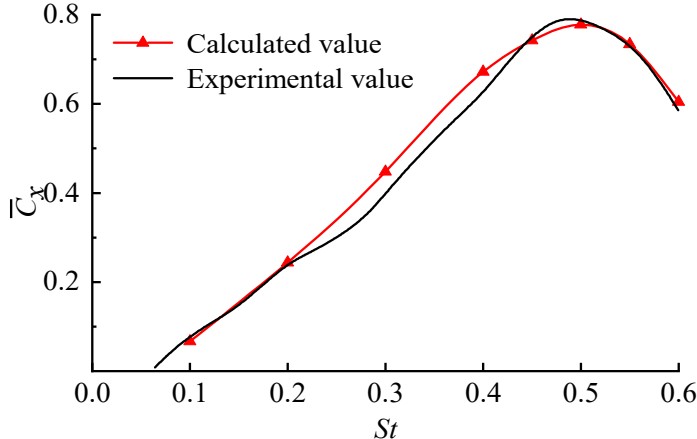

**Figure 5.** Comparison between numerical simulation results and experimental data in the literature [24].

It can be seen from Figure 5 that the simulation calculation value of average thrust coefficient of flapping hydrofoil is basically consistent with the experimental results obtained in MIT drag tank laboratory, which proves that the numerical calculation method used in this paper is correct and effective.

### 2.2.4. Computational Domain and Grid Generation

To study the influence of passage structure on flapping hydrofoil, it was necessary to fully develop the flow field through the hydrofoil flapping to reduce the influence of the wall surface on the simulation results. The length of the basin behind the flapping hydrofoil is 20 c, that is, the total length in the x direction is 8 m, and the length in the positive x direction is 6m. The initial rotation center of the flapping hydrofoil is located at the origin. In addition, to save computing costs, the computing domain was divided into unstructured grid regions centered at the origin and 1m long in the x direction, and the hydrofoil movement was realized with the help of dynamic grid technology. In addition, to better capture the flow field of the airfoil and the wall, the boundary layer was divided around it, and the thickness of the first layer grid was 0.0015 m (y +> 37.5). The boundary layer of the airfoil and its several outer layers of grids were separately divided into a fixed region, which does not participate in grid reconstruction when the hydrofoil moves, to ensure the grid quality around the airfoil. The rest areas were divided by a structured grid. The computational domain and grid division without pitch angle at the initial time is shown in Figure 6.

In grid independence analysis, the number of different grids is used for numerical simulation. The results show that when the number of grids reaches 83,000, its influence on the calculation results can be ignored. To reduce the calculation time, 82,994 grids are selected.

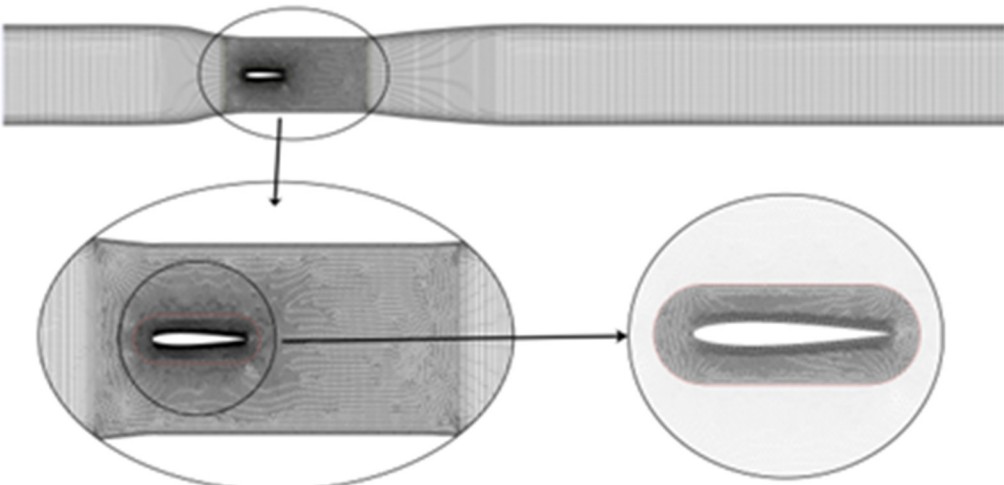

**Figure 6.** Computational domain and grid partition.

### 2.3. Experimental Setup and Uncertainty Analysis

### 2.3.1. Flapping Hydrofoil Bionic Pump

The flow-flapping hydrofoil bionic pump includes four flow parts: inlet passage, water flapping wings, contraction expansion passage, and the outlet passage. The chord length of the hydrofoil was 300 mm, the initial angle of attack was 30°, and the clearance between the spanwise direction of the hydrofoil and both sides of the flow passage was 5 mm. The three-dimensional entity diagram of the flapping hydrofoil water-pushing device is shown in Figure 7.

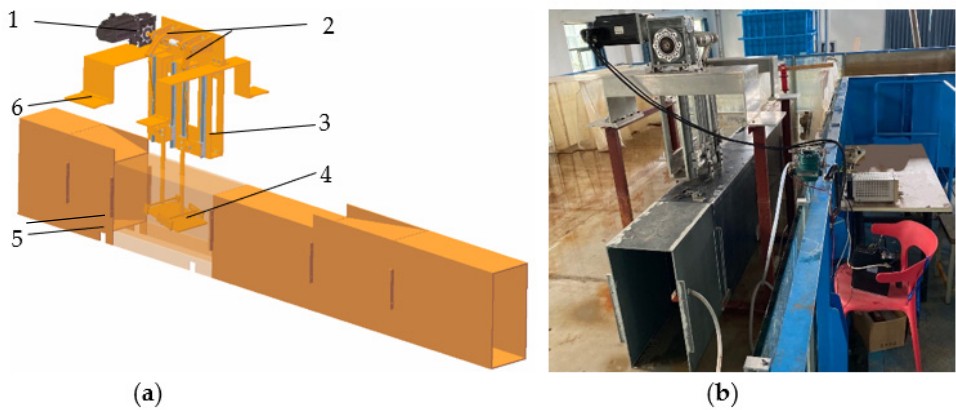

**Figure 7.** (**a**) Schematic diagram of flapping hydrofoil water pushing device test bench: 1. Motor and reducer; 2. Double crank mechanism; 3. Slide rail; 4. Flapping wing; 5. Contraction and expansion passage; 6. Support frame (**b**) Photograph of the flapping hydrofoil water-pushing device test bench.

To verify the water-pushing performance of the flapping hydrofoil device, a flapping hydrofoil test device was designed, as shown in Figure 7a. The Servo motor and gearbox provide the power required for device operation; the slider crank mechanism converts the rotary motion into linear motion, driving the blade to perform the heave motion, and at the same time, it coordinates the phase difference between two groups of slider-crank mechanisms to achieve the pitch motion of the blade, to achieve the sinusoidal flutter of the blade.

As shown in Figure 7b, the device was erected above the 4 m × 0.5 m × 0.7 m open passage, and the probe of WIM-@ADV acoustic Doppler current meter was located at the outlet of the passage, which was used to measure the velocity of each point in the outlet section, with a measuring range of 0~3 m/s, a measuring accuracy of 0.005 m/s and a sampling frequency of 50 Hz. The inlet and outlet side plates of the flow passage were

provided with pressure leading ports, which were externally connected to the differential pressure sensor through rubber tubes, with a measuring range of 300 pa and a minimum scale of 0.2 pa. The input power was measured by electrical measurement.

2.3.2. Uncertainty Analysis of Experiment

Efficiency synthesis error [25] is the square sum root of systematic error and random error:

$$E_\eta = \pm\sqrt{E_{\eta \cdot S}^2 + E_{\eta \cdot R}^2} \tag{11}$$

The total system error of the efficiency of the pump device performance test of the test bench is the root and square of the individual system errors:

$$E_{\eta,S} = \pm\sqrt{E_{Q \cdot S}^2 + E_{H \cdot S}^2 + E_{P \cdot S}^2} \tag{12}$$

where $E_{Q \cdot S}$ is systematic error of the Doppler velocimeter, $E_{H \cdot S}$ is systematic error of the differential pressure transmitter, and $E_{P \cdot S}$ is systematic error of the clamp power meter, these systematic error parameters are dependent on testing equipment shown in Table 2.

**Table 2.** Passage structure parameter table.

| Terms | Equipment | Type | Systematic Error |
|---|---|---|---|
| Flow | Doppler velocimeter | WIM-@ADV | ±1% |
| Head | Differential pressure | 3051 | ±0.2% |
| Current | Clamp power meter | VC6412D | ±2.5% |

The total random error can be calculated with the following equation:

$$E_{\eta,R} = \pm\sqrt{E_{Q \cdot R}^2 + E_{H \cdot R}^2 + E_{P \cdot R}^2} \tag{13}$$

where $E_{Q \cdot R}$ is the random error of flow testing, $E_{H \cdot R}$ is the random error of head testing, $E_{P \cdot R}$ is the random error of torque speed testing.

The systematic error was estimated based on the systematic error of each testing equipment and the previous test experience. The total random error was calculated by the method of probability statistics based on the test data of this pumping system model (Doppler velocimeter: 2%, Differential pressure: 0.98%, Clamp power meter: 3%). The total uncertainty $E_\eta$ was ± 4.61%.

In order to verify the reliability of the test bench, the bionic pump devices with 0.6m width passage and different flapping frequencies from 0.1 Hz~0.7 Hz were tested repeatedly, under the same test method and operating conditions. The test results are shown in Table 3. With the increase of frequency, the speed change trend of the bionic pump device in the three tests is basically the same, and the data collected under similar working conditions are very close, indicating the reliability of the test results.

**Table 3.** Performance repeatability test of bionic pump device.

| | *f*/Hz | 0.1 | 0.2 | 0.3 | 0.4 | 0.5 | 0.6 | 0.7 |
|---|---|---|---|---|---|---|---|---|
| | Expeiment1 | 0.041 | 0.079 | 0.122 | 0.159 | 0.196 | 0.234 | 0.259 |
| v/m·s$^{-1}$ | Expeiment2 | 0.043 | 0.082 | 0.128 | 0.148 | 0.197 | 0.238 | 0.262 |
| | Expeiment3 | 0.049 | 0.085 | 0.132 | 0.160 | 0.201 | 0.240 | 0.268 |

## 3. Results

### 3.1. Influence of Passage on the Dynamic Performance of the Water-Pushing Device

The hydrodynamic performance of the water-pushing device is affected by the passage width, which is reflected in the instantaneous thrust and lift coefficient of the flapping wing. As shown in Figure 8, four typical groups are selected for comparative analysis.

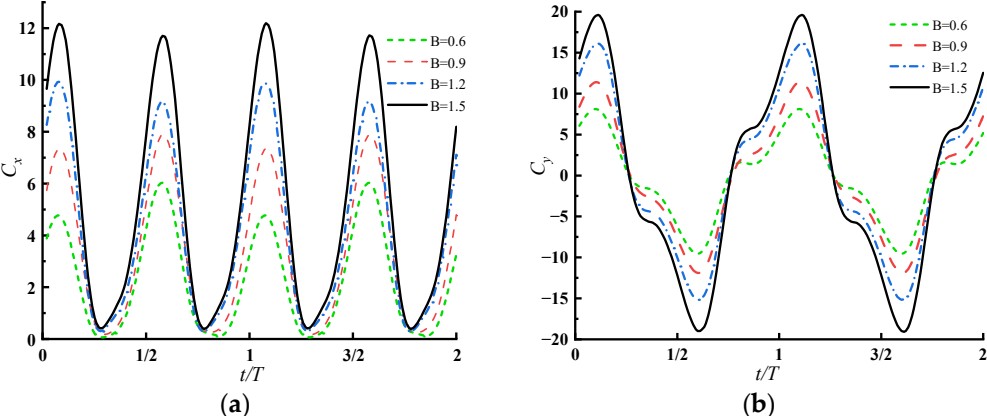

**Figure 8.** Variation of instantaneous thrust and lift coefficients with different passage width: (**a**) Instantaneous thrust coefficients; (**b**) Instantaneous lift coefficients.

In order to facilitate comparison, the relative values of dimensionless time of each period are taken as abscissa. It can be seen from Figure 8a that when the flapping hydrofoil starts flapping upward from the equilibrium position, the instantaneous thrust coefficient first increases and then decreases. When flapping reaches the maximum flapping amplitude, the instantaneous thrust coefficient reaches the valley point, then the flapping hydrofoil starts flapping backward, the instantaneous thrust coefficient starts to increase again, and quickly reaches another peak after passing the equilibrium position. Moreover, the instantaneous thrust coefficient is always greater than zero in a movement period, that is, flapping hydrofoil always promotes the flow in the whole flapping process. According to Figure 8b, the instantaneous lift curve is basically distributed symmetrically along the axis, and the average lift coefficient under all working conditions is close to zero. The maximum values of flapping wing thrust and lift coefficient increase significantly with the increase of passage width.

To study the influence of flow width on the hydrodynamic performance of a flapping hydrofoil device, this paper selects 10 groups of numerical simulation calculations with the flow passage width gradually increasing from 0.6 m to 1.5 m and changing every 0.1 m. The results are shown in Figure 9.

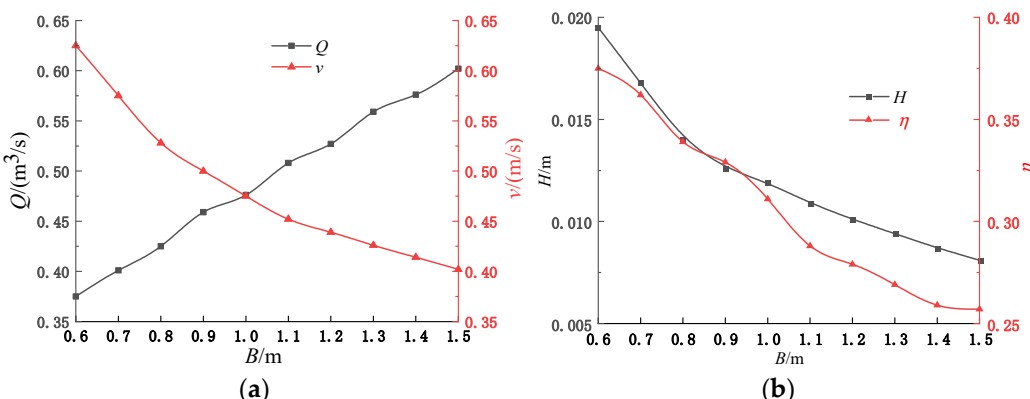

**Figure 9.** Comparison of hydrodynamic performance of different passage widths: (**a**) Width-flow and velocity curve comparison; (**b**) Width-head and efficiency curve comparison.

By observing Figure 9a, it can be found that with the increase in passage width, the flow velocity at the outlet section will decrease, the passage width is increased from 0.6 to 1.5 m, and the average velocity at the exit is reduced from 0.625 m/s to 0.40 m/s, but the flow of the flapping hydrofoil device can still keep increasing. Pump water flow increased from 0.375 $m^3$/s to 0.60 $m^3$/s, because the influence of the decrease of flow velocity is smaller than that of the increase of passage width. In addition, it can be seen from Figure 9b

that the average head of the flapping hydrofoil changes with the passage width, that the average head of the flapping hydrofoil device decreases with the passage width, and the average head of the flapping hydrofoil is less than 0.02m within the calculation range, which well meets the requirements of flapping hydrofoil devices for large flow and low head. The efficiency of the flapping hydrofoil device decreases gradually with the increase of passage width. When the passage width is 0.6 m, the water pumping efficiency of the flapping hydrofoil device has a maximum value of 37.4%, and the efficiency gap between 0.6 m and 1.5 m is as high as 11%, while the efficiency of traditional axial flow pump is less than 30% when the head is less than 1 m.

The passage width has a great impact on the pumping performance of the flapping wing hydrodynamic device. With the increase of passage width, the average flow of the device increases steadily while the average head decreases, which can better adapt to the water conveyance requirements of low head and large flow in plain rivers. To further analyze the reasons for the influence of passage width on the pumping performance of flapping hydrofoil, four groups of simulation results with passage widths of 0.6 m, 0.9 m, 1.2 m, and 1.5 m were selected to draw the velocity and vorticity nephogram of flapping hydrofoil under different passage widths, as shown in Figures 10 and 11.

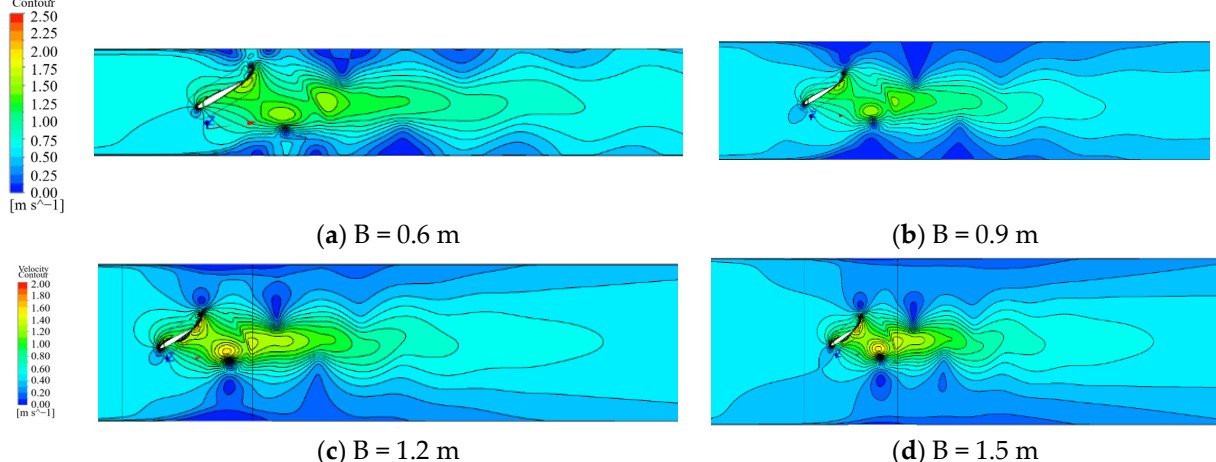

(**a**) B = 0.6 m      (**b**) B = 0.9 m

(**c**) B = 1.2 m      (**d**) B = 1.5 m

**Figure 10.** Velocity cloud chart of flapping hydrofoil under different passage widths.

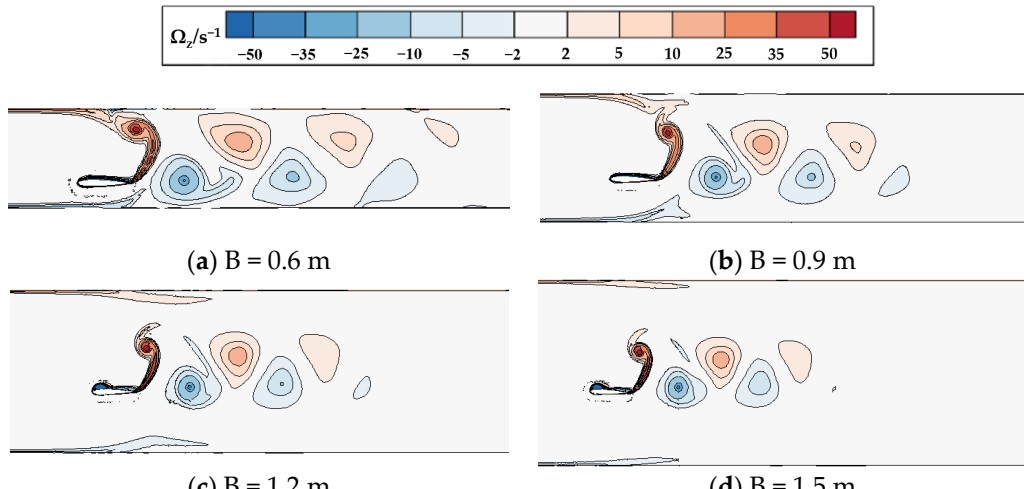

(**a**) B = 0.6 m      (**b**) B = 0.9 m

(**c**) B = 1.2 m      (**d**) B = 1.5 m

**Figure 11.** Vorticity cloud maps of flapping hydrofoils under different passage widths.

It can be seen from Figure 10 that during the continuous increase of passage width, the movement form of the flapping hydrofoil remains unchanged, the range of water body directly affected remains unchanged, and the high-speed jet formed by the flapping

hydrofoil also remains unchanged. Because of the influence of the wake on the water body near the wall, the water bodies on both sides are in a low-speed state except for the central jet area, and the overall velocity of the water body decreases with the increase of the passage width. However, in the process of the continuous backward development of the wake, the center jet continuously drives the low-speed fluid on both sides, and finally can form a relatively stable wake. Although the overall flow rate decreases with the increase of the passage width, the increased outlet cross section brings a huge flow increase, and the flow of the flapping wing hydrodynamic device still increases with the increase of the passage width.

Select the component $\Omega_z$ of vorticity in z direction as the characteristic quantity, and then draw the vorticity contour map under various working conditions. The calculation formula of the vorticity component in the z direction [26] is:

$$\Omega_z = \frac{\partial u_y}{\partial x} - \frac{\partial u_x}{\partial y} \tag{14}$$

where $\Omega_z$ is the component in $z$ direction, $u_x$, $u_y$ is the velocity in $x$, $y$ direction.

Observing the vorticity nephogram of flapping hydrofoil under different passage widths in Figure 11, it can be found that the wake vortex of flapping hydrofoil hydrodynamic device presents a regular double row anti-Karman vortex street shape regardless of the channel width. Moreover, when the channel width is 0.6 m, the interaction between the wake and the wall will cause the shape of the wake to flatten due to the narrow passage width, the shape of the wake of the flapping wing hydrodynamic device is basically the same under the other three channel widths, but the dissipation speed of the wake accelerates with the increase of the channel width.

By comparing and observing the first vortex after flapping hydrofoil, it can be found that the size of the vortex generated from flapping hydrofoil is basically the same, which can completely cover the 0.6 m-wide passage but can only cover about 1/2 of the 1.5 m wide passage section. This gives the water body in the narrow passage a high velocity after passing the flapping hydrofoil, while some areas on both sides of the wide passage are less affected by the flapping hydrofoil, but the overall velocity is low due to the effect of the wake vortex. When the flow passage is narrow, the energy is mainly converted from kinetic energy to pressure energy during the downward development of the wake, and the dissipation of the wake is relatively slow, and the loss is small. When the flow passage is wide, the energy of the water body after passing the flapping hydrofoil is mainly distributed in the wake vortex. With the development of the downstream, only the kinetic energy of some vortices is converted into pressure energy, and the rest of the energy is transferred outward with the process of the wake vortex dissipation, which drives the low-speed fluid around through the interaction between the fluids. The energy loss is large, leading to the flapping wing hydrodynamic device pushing the water body in a larger passage width; although it has a high flow rate, the average flow rate, average head, and efficiency decrease.

### 3.2. The Influence of Flow Passage Structure on Hydrofoil Propulsion Characteristics

To further analyze the influence of flow passage structure on flapping hydrofoil pumping performance, the curve of average thrust and lift coefficient of flapping hydrofoil hydrodynamic device changing with flow passage structure is drawn as shown in Figure 12.

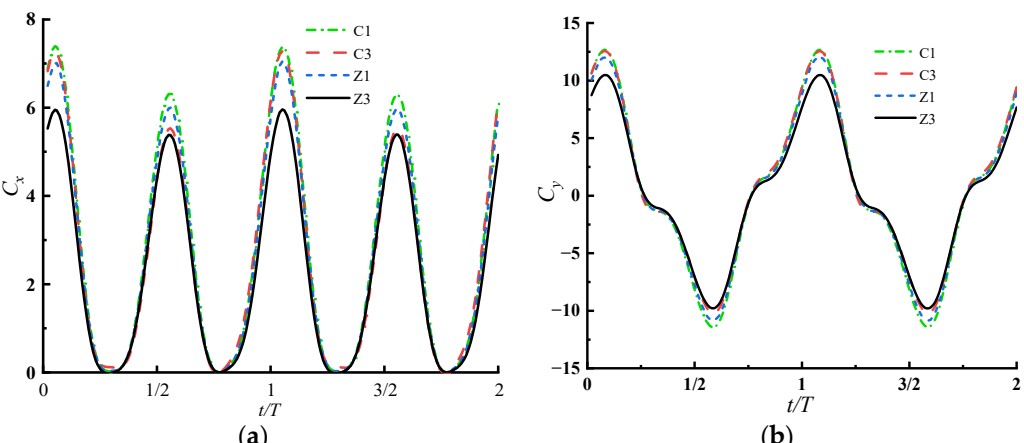

**Figure 12.** Variation of instantaneous thrust and lift coefficients with different passage structures: (**a**) Instantaneous thrust coefficients; (**b**) Instantaneous lift coefficients.

By comparing Figure 8, it can be seen that the contraction expansion passage structure will reduce the average thrust and coefficients of the hydrofoil. It can be seen from the data of Group C or Group Z separately that the average thrust and lift coefficients of the flapping hydrofoil will decrease with the increase of the length L of the central contraction section.

Flapping hydrofoil hydrodynamic device has a different hydrodynamic performance due to the change in passage width. The main reason for the low efficiency of the wide passage is that the airfoil has less water directly acting on it. Therefore, this paper designed a variety of passage schemes as shown in Figure 4, to combine the advantages of wide and narrow passages to obtain high efficiency and high flow.

Figure 13a shows the average flow velocity and head change curve of the flapping wing hydrodynamic device under different flow passage structures. By observing the three groups of data in Group C and Group Z, respectively, it can be found that the increase of the length L of the central contraction section can effectively improve the average flow velocity and head of the flapping wing hydrodynamic device, when the passage structure is Z3, the flow rate is as high as 0.4376 m$^3$/s, and the corresponding head is 0.016 m, which is obviously improved compared with the simple narrow passage. Moreover, the elevation of the right angle transition is more obvious, which indicates that the length L of the central contraction section is the main influence part in the contraction expansion passage.

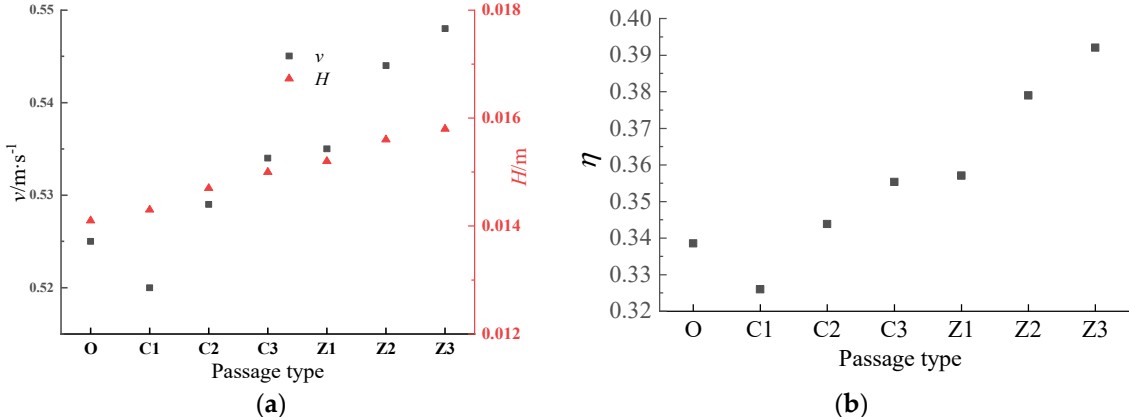

**Figure 13.** Comparison of hydrodynamic performance of different passage structures: (**a**) Type-velocity and head curve comparison; (**b**) Type-efficiency curve comparison.

Figure 13b shows the efficiency change curve of the flapping wing hydrodynamic device under different passage structures. By observing the changes in the average efficiency of Group C and Group Z, respectively, it can be found that the pumping efficiency of the

flapping wing hydrodynamic device has been improved with the increase of the length L of the central contraction section. Within the simulation scope, the Z3 group with the length L of the central contraction section = 2 m has the maximum pumping efficiency of 39.2%, which is 5.4% higher than the 33.9% of the rectangular passage. Compared with the pump efficiency of 37.4% when the passage width is 0.6 m, the pump efficiency is also improved by 1.8%. By the same observation of Group C1, it can be found that the simple contraction and expansion of the pipeline will not improve the water pumping efficiency of the flapping wing hydrodynamic device but will hinder it to a certain extent.

Both contraction and expansion pipes can effectively improve the pumping performance of flapping wing hydrodynamic devices, and the length L of the central contraction section is the main factor affecting the pumping performance of the contraction and expansion pipes. Although the simple contraction expansion structure can improve the average head of the device, it reduces the efficiency and flow of the device and has no significant effect on improving the pumping performance of the flapping wing hydrodynamic device. To further analyze the efficiency change rule of the flapping wing hydrodynamic devices in different passage structures, after the outlet velocity of the flow field is stable, at 3/4T of the fifth flapping cycle. the vorticity cloud charts in different passage structures (C1,C3,Z1,Z3) are drawn, as shown in Figure 14.

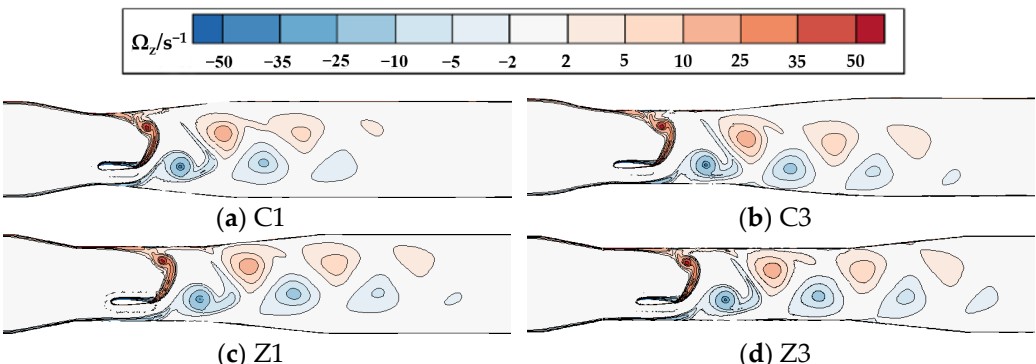

**Figure 14.** Vorticity cloud chart with different flow passage structure.

Based on the comprehensive observation of Figure 14, it is found that no matter what the flow passage structure is, the wake presents a regular double-row anti-Karman vortex street, and the wake will have a certain loss through the expansion pipe. By comparing and observing the first vortex after the flapping hydrofoil, it can be found that the vorticity of the first vortex under the four flow passage structures is the same, that is, the energy obtained by the tail vortex during flapping hydrofoil movement is the same. However, the dissipation speed of the wake decreases with the increase of the length L of the central contraction section. This is because the water body in the narrow passage is affected by the wake, and the overall flow rate is high. During the backward development of the wake, the energy is mainly converted from kinetic energy to pressure energy, and the loss is small. When the wake enters the expansion pipe, it also needs to drive the low-speed fluid on both sides when the velocity decreases, and the energy loss is obvious in the process of transfer. When the length L of the central contraction section is small, the tail vortex keeps this higher energy flowing into the expansion tube and violently interacts with the low-speed fluid on both sides. A large amount of energy is lost in this process, so the water pumping efficiency of the flapping wing hydrodynamic device in the C1 flow passage is low. The length L of the central contraction section is larger, and the fluid flow pattern in the wake is more uniform, the fluid interaction is less after entering the expansion pipe, the energy loss is smaller, and the water pumping efficiency of the flapping wing hydrodynamic device is higher.

### 3.3. Performance Test

The spread length of the test blade is 0.3 m, the test frequency is 0.1~0.7 Hz, and it changes once every 0.1 Hz. The acoustic Doppler velocimeter is used to measure the velocity at six measuring points uniformly set at the outlet of the square passage. The single measurement shall be not less than 12 cycles. The average value of the multiple measurements shall be taken as the velocity at the measuring point, and the average velocity of the six measuring points shall be taken as the plane velocity.

Compare the test flow rate with the simulation results, as shown in Figure 15. When the frequency is 0.7Hz, the experimental flow rate is 0.252 m/s. The overall trend of the experimental and simulation results is the same. The flow rate of the device increases with the increase of the motion frequency. Compared with the rectangular passage, the contraction passage can effectively improve the flow rate of the water-pushing device. However, the simulation result is slightly higher than the test result, and the main reason for this difference is that there are inevitably some gaps at the joint of the test flow passage, which leads to the leakage of some pressurized water bodies during the test process, indirectly losing some energy; there is also a certain distance between the flapping blade and the flow passage, leading to the generation of three-dimensional effect, which has a certain impact on the test results.

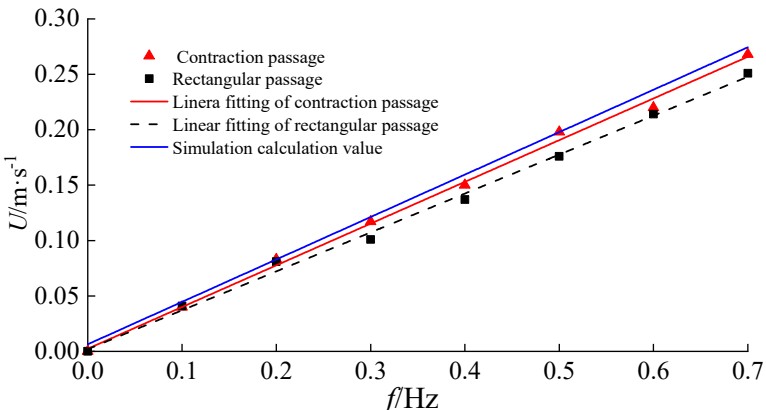

**Figure 15.** Comparison of flow velocity between test and simulation.

The results of the test flow field taken with a high-speed camera are shown in Figure 16. Due to the fast diffusion of ink, only one wake exists at each time. It can be found that the test flow field is the same as the simulation flow field, which confirms the correctness of the simulation results from the side.

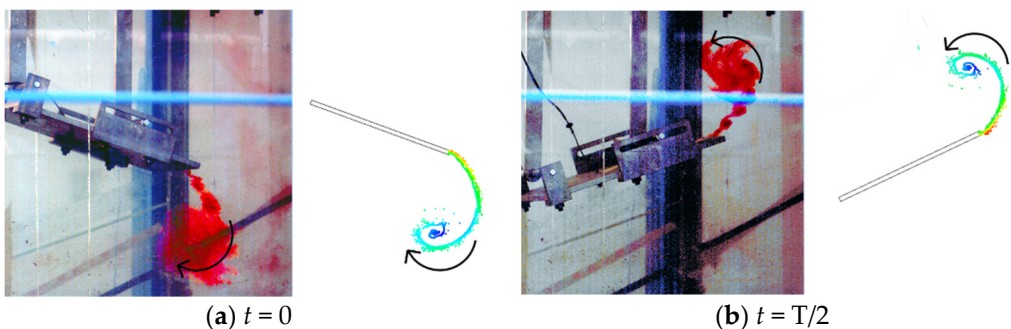

(**a**) *t* = 0             (**b**) *t* = T/2

**Figure 16.** Flow field diagrams of test and simulation.

### 4. Conclusions

The research group proposed a new water body propulsion mode of flapping hydrofoil applied to ultra-low lift water body propulsion, which can meet the application conditions of ultra-low lift and large flow in plain small rivers and other areas, and effectively broaden

the high efficiency range of the device in this application situation. This paper considers the wall effect, innovatively combines flapping wing with flow channel, and explores the law of the influence of different flow channel structures on the water performance of a flapping wing pump through numerical simulation and test methods, which has important engineering significance. The following conclusions are obtained:

(1) In a certain range, the thrust coefficient of the flapping wing is positively related to the channel width, while the thrust coefficient of the flapping wing will be reduced by using the contraction expansion passage, and the trend is more obvious with the increase of the center contraction length. In addition, on the premise of constraining the center shrinkage length, the thrust coefficient of Z1 group with right angle transition is higher than that of C3 group with fillet transition.

(2) When other parameters are fixed, increasing the passage width can effectively increase the average flow of the flapping wing hydrodynamic device, but its head, flow rate, and pumping efficiency are reduced to a certain extent. When the passage width is 0.6 m, the pumping efficiency of the device is 37.4%, which is 11.2% higher than that when the passage width is 1.5 m, but the flow rate is only 0.375 m$^3$/s, which is far lower than 0.605 m$^3$/s when the passage width is 1.5 m.

(3) Under the same working conditions, the expansion and contraction passages can effectively improve the pumping performance of the flapping wing hydrodynamic device. In addition, the length of the central contraction section in the contraction expansion passage is the main factor affecting the pumping performance of the flapping wing hydrodynamic device. The flow, velocity, and efficiency of the flapping wing hydrodynamic device increase with the length of the central contraction section. When the length of the contraction section is 2 m, the pumping efficiency of the device is as high as 39.2%, and the flow rate is as high as 0.4376 m$^3$/s, and the head is 0.016 m. Compared with the simple narrow flow passage, it also shows obvious improvement, and meets the application design concept of large flow and high efficiency under the ultra-low head condition.

**Author Contributions:** E.H. presented the main idea of analyzing the performance of the flapping wing water pump under different passage structures; H.L. contributed to the overall composition and writing of the manuscript; W.C. conducted numerical theory research; R.X. reviewed the manuscript; S.T. provided test prototype; D.J. revised the manuscript. All authors have read and agreed to the published version of the manuscript.

**Funding:** This research was funded by the National Natural Science Foundation of China (Grant No. 51976202, 61772469) and the Zhejiang Provincial Key Research and Development Project (Grant No.2021C03019) and the Zhejiang Provincial Key Laboratory of Rural Water Conservancy and Hydropower Resources Allocation and Control Key Technologies Open Fund Project (Grant No. UZJWEU-RWM-20200304B).

**Institutional Review Board Statement:** Not applicable.

**Informed Consent Statement:** Not applicable.

**Data Availability Statement:** Not applicable.

**Conflicts of Interest:** The authors declare no conflict of interest.

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
