# Peer review of "Investigation on the Influence of Flow Passage Structure on the Performance of Bionic Pumps"

_processes, doi:10.3390/pr10122569_

Round 1
Reviewer 1 Report
The paper discusses the performance of Flapping Hydrofoil Bionic pump under different passage structures of the river. The paper is well written and can be considered for possible publication subjected to the revision based on the following queries.
1. The motions such as “heave” and “pitch” may be explained in Figure2.
2. The fonts may be enlarged in Figure 1.
3. What is the expression for the average velocity at outlet ?
4. The captions of Figure 1, 2 and 3 appear to be same. The captions may be suitably modified to bring out the essence of the Figures.
5. The flow passages (c, z, and o) may be clearly shown in a different figure to highlight the differences in their features that affect the performance of the flaps.
6. The updating of dynamic mesh is not clear in Figure (4)
I. The co-ordinate axes must be shown.
II. Passage type should be mentioned.
III. “Four main points” should be elaborated.
7. The experimental setup can be explained with detailed instrumentation along with the positioning of the sensors.
8. A domain independence test of the CFD analysis may be presented.
9. What are the sources of systematic error? The authors may explain this error has been obtained for a sample calculation of head, flowrate and efficiency may be shown with data for calculating, , lift coefficients, Strouhal no, etc.
10. How are the random errors obtained?
11. In Figure 9, Q and V plots should be clearly depicted.
12. What are the effects of hydrofoil shape on vortex generation?
13. In Figure 13, the points on the plot should not be joined as a continuous plot. Also, the effects of each of the passages on the performance of the flapping hydrofoil should be explained to come to the conclusion of #3 as mentioned in section 4.
14. In the Figure 14.
I. The passage structures should be clearly marked with the lengths of expansion and contraction sections.
II. The position of the flap and the time of the flapping cycle should be mentioned.
15. Section 3.3 may be used as the validation of the numerical model developed.
16. What are the design decisions recommended from the present study?
Author Response
请参阅附件

Reviewer 2 Report
Reviewer Report
Recommendation: Minor Revision
Reviewer's comments:
Paper is of current interest and falls in the scope of the journal, however, there are the following suggestions authors should address and then I welcome for publication:
Comment 1: Title of the paper should be modified.
Comment 2: Improve the quality of the paper by fixing some typos errors.
Comment 3: Add the novelty of our work in the last paragraph of the introduction.
Comment 4 Improve the quality of the Figure 1 and Figure 2 and Figure 3.
Comment 5: Results and discussion part is week. Improve it by giving physical meaning
Comment 6: The originality of the paper needs to be stated clearly. It is of importance to have sufficient results to justify the novelty of a high-quality journal paper.
Comment 7: Finally, applications of the current investigation have to be highlighted.
Reviewer 3 Report
The article submitted for review has a high scientific and practical value. The reviewed article deals with the topic influence of flow passage structure on the hydrodynamic performance of flapping hydrofoil bionic pump. The authors concentrated on the main working component of flapping wing bionic pump that is NACA0012 airfoil. In this paper, the two-dimensional flow passage models are established, and the flapping of the airfoil is realized by using the dynamic grid technology. Based on the continuity equation, k-ε turbulence model, and Reynolds time-averaged equation, the flapping hydrofoil device is simulated by transient calculation. The hydraulic performance characteristics of various passages with different widths, such as square passages, micro-arc passages, and convergent-divergent passages, are calculated and simulated. In addition, the authors present in their work compared the simulation results with the experimental results. The work presented for review is very interesting, deals with an interesting topic and fits well with the scope of the journal. The research methodology presented is good. The conclusions are rightly worded. The research results and their analysis are also adequately presented. However, the paper contains limited discussions, especially since there is a lack of comparison to the results of other authors, therefore a significant discussion regarding the underlying mechanisms controlling the observed findings should be realized. The bibliographic list consists of most of the publications by authors from the PRC, which shows a certain degree of bias in the selection of sources. It is required either to revise the bibliographic list, or to expand the list by adding publications of authors from other countries to it.
Round 2
Reviewer 1 Report
The paper has been improved from its earlier version and is recommended for possible publications
Reviewer 3 Report
The article has been corrected and is suitable for publications